# Synergistically Enhanced Electrochemical Sensing of Food Adulterant in Milk Sample at Erbium Vanadate/Graphitic Carbon Nitride Composite

**DOI:** 10.3390/s24061808

**Published:** 2024-03-11

**Authors:** U. G. Anushka Sanjeewani, Sea-Fue Wang

**Affiliations:** Department of Materials and Mineral Resources Engineering, National Taipei University of Technology, No. 1, Section 3, Chung-Hsiao East Road, Taipei 106, Taiwan

**Keywords:** ErVO_4_, g-C_3_N_4_, composite, electrochemical detection, dimetridazole

## Abstract

Dimetridazole (DMZ), a nitroimidazole derivative, is a notable antibiotic that has garnered growing interest in the medical community owing to its noteworthy pharmacological and toxicological properties. Increasing interest is being directed toward developing high-performance sensors for continuous monitoring of DMZ in food samples. This research investigated an electrochemical sensor-based nano-sized ErVO_4_ attached to a sheet-like g-CN-coated glassy carbon electrode to determine dimetridazole (DMZ). The chemical structure and morphological characterization of synthesized ErVO_4_@g-CN were analyzed with XRD, FTIR, TEM, and EDS. Irregular shapes of ErVO_4_ nanoparticles are approximately 15 nm. Cyclic voltammetry (CV) and differential pulse voltammetry (DPV) were followed to examine the electrochemical performance in pH 7 phosphate buffer solution for higher performance. This electrochemical sensor showed a low detection limit (LOD) of 1 nM over a wide linear range of 0.5 to 863.5 µM. Also, selectivity, stability, repeatability, and reproducibility studies were investigated. Furthermore, this electrochemical sensor was applied to real-time milk sample analysis for the detection of analytes.

## 1. Introduction

1,2-Dimethyl-5-nitro imidazole (dimetridazole or dimet or DMZ) is a nitroimidazole-type veterinary drug used for bacterial and protozoal infections in animals, and it is used as a growth promoter in meat production to enhance efficiency in feeding. DMZ is banned in food processing in many countries because the residue of DMZ leads to mutagenicity, carcinogenicity, and genotoxicity in the human body. While the United States, Canada, Japan, and China have already banned DMZ, the United States and the European Union have also established a maximum residue limit of biological samples such as meat and milk. Among different biological matrices, milk has a lower maximum residue limit, and the detection limit is required to be ≤2 μg/kg [1,2,3,4,5].

DMZ can be determined by electrochemical methods, immunoassay, thin-layer chromatography (TLC), gas chromatography (GC), gas chromatography–mass spectrometry (GC–MS), high-performance liquid chromatography (HPLC), and liquid chromatography–mass spectrometry (LC–MS). Among these methods, GC and GC–MS need derivatized analytes, and HPLC needs a large sample volume, a long elution time, and is of high cost [6,7]. Because of these drawbacks in chromatography methods, the electrochemical method with a low detection limit becomes compatible with detecting DMZ in different environments. The advantages of electrochemical methods are low cost, high sensitivity, and detection of a wide range of target analytes. Cyclic voltammetry is a successive electrochemical method to analyze the oxidation and reduction reaction of a substance [8]. Coating nanomaterials can increase sensitivity, absorption, adsorption, and efficiency of application. Large specific surface areas of nanomaterials help to develop better limits of detection (LOD) and sensitivity [9].

Oxide-based materials are frequently used in electrochemical applications. Rare earth elements have gotten more attention in recent years because rare earth oxides are used in photocatalysis, thin film phosphors, and magnetics technologies. Among these oxides is rare earth orthovanadate (RVO_4_), which has unique chemical and physical properties. Moreover, rare earth orthovanadate structure exists in monoclinic monazite-type and a tetragonal zircon-type polymorphous compounds [10]. Erbium (Er) is the 11th atom of REE, attracting attention in recent doping photocatalyst applications [11]. Vanadium has oxidation states of +5, +4, +3, and +2 [12]. Diverse oxidation numbers of vanadium can be mixed with varied chemical elements, opening access to use in many applications. The study of the electrocatalytic behavior of erbium vanadate (ErVO_4_) is promising due to the compound’s high stability and excellent repeatability value. Moreover, RVO_4_; R = La, Ce, Nd, Sm, Eu, and Gd are the frequently used elements in electrochemical sensor applications and there are fewer electrochemical studies have been reported on ErVO_4_ [13,14].

Hydrothermal, solvothermal, sol-gel, Pechini, co-precipitation, and ion exchange methods can synthesize nano-sized materials. Furthermore, chemical methods have more advantages than mechanical methods. Even high-temperature methods are available, but some nanomaterials are unstable at elevated temperatures, which is a disadvantage for commercial usage [15,16]. The hydrothermal method uses a temperature range of 60–200 °C and controls the nucleation and crystal growth of RVO_4_. Therefore, effective nano-sized powder with a high surface area of ErVO_4_ can be easily prepared from the hydrothermal method for sensor application; catalytic performance depends on particle size and surface area [17,18,19]. 

Carbon-based materials are widely studied due to the advantages of structural chemical properties and electrical characteristics. Many researchers have proven that doping heteroatoms, such as nitrogen, sulfur, boron, and iodine, with carbon can enhance electrochemical performance. Among previously mentioned heteroatoms, nitrogen attached to carbon studies show excellent catalytic activity. Graphitic nitride-based nanocomposites are low cost, have low toxicity, are highly thermal durable, and can be applied in photocatalytic applications, biomedical applications, and hydrogen generation for the reasons mentioned previously. Thermal treatment with melamine, dicyanamide, and cyanamide can be applied in order to synthesize g-CN; melamine is prominent for having a large g-CN surface area. High electrical conductivity due to a lone pair of nitrogen atoms attached to carbon band gap of ~2.7 eV is beneficial to electrochemical sensor usage as carbon support with ErVO_4_. Furthermore, the overall detecting properties are improved by molecular-level interactions and size confinement effects as a result of the improved synergistic effects between g-CN and ErVO_4_ as well as the significantly altered active surface area of the ErVO_4_@g-CN nanocomposite [20,21,22,23,24,25,26,27,28,29,30,31]. 

In this research work, erbium vanadate attached to a graphitic carbon (ErVO_4_@g-CN) heterostructure was prepared by sonication, and optimization for the electrochemical detection of DMZ was carried out. The proposed ErVO_4_@g-CN/GCE-modified electrode enhanced the electrochemical behaviors on DMZ determination with a wide linear range and low detection limit. Furthermore, the ErVO_4_@g-CN/GCE-modified electrode revealed good selectivity, stability, repeatability, and reproducibility with a less than ±5% relative standard deviation. Furthermore, the practicability of the proposed sensor was applied to spiked milk samples. Finally, the above-discussed electrochemical sensor is a promising candidate for electrochemical monitoring of DMZ in real-world samples.

## 2. Experimental Section

### 2.1. Materials and Reagents

Erbium (III) nitrate pentahydrate (ErH_10_N_3_O_14_)_,_ ammonium metavanadate (NH_4_VO_3_ ≥ 99% purity), and urea (CH_4_N_2_O) were purchased from Sigma Aldrich, China and Showa Chemical Co. Ltd., Tokyo, Japan. Potassium chloride (KCl), Potassium ferricyanide (K_3_Fe(CN)_6_), and Potassium ferrocyanide (K_4_Fe (CN)_6_.3H_2_O) were obtained from Showa Chemical Co. Ltd., Tokyo, Japan. Sodium phosphate dibasic and sodium dihydrogen phosphate (Na_2_HPO_4_ and NaH_2_PO_4_) were utilized to prepare 0.1 M (pH 7) PBS (phosphate buffer solution). All the electrochemical experiments were carried out using 0.1 M PBS (pH 7) as the supporting electrolyte. Milk samples were brought in from a nearby convenience store. These samples were centrifuged and diluted before further experiment. Prepared samples were spiked with known concentrations to use for the detection of DMZ separately.

All analytical grade chemicals were used as received without further purification in these experiments. Details regarding materials characterization instruments are given in supporting information.

### 2.2. Intruments

XRD analysis was performed using a Bruker, Rigaku D/maxB, DMX-2200 instrument, Taiwan. The morphology and elemental mapping of sample was analyzed using a high-resolution (HR) transmission electron microscope (TEM) (JEOL JEM-2100F (HR)) operating at 200 kV and by energy-dispersive X-ray spectroscopy using EDAX AMETEK Inc., DigitalMicrograph^®^ software. Fourier transform infrared spectra (FTIR) were tested using Nicolet 6700 (Thermo Fisher Scientific, USA). Electrochemical impedance spectroscopy and electrochemical measurements were carried out with Ω-metrohm auto lab (AUT51770, 100–240 V~75 VA50/60 Hz) instrument, Taiwan (Nova 2.1 software) and CHI 1211c electrochemical workstation (Taiwan). A glassy carbon electrode (surface area = 0.072 cm^2^), Ag/AgCl, and a Pt wire were the working, reference, and counter electrodes, respectively in this electrochemical analysis.

### 2.3. Synthesis of ErVO_4_

The hydrothermal process was followed in synthesizing ErVO_4_ in order to obtain nanocrystalline material. Erbium (III) nitrate pentahydrate (ErH_10_N_3_O_14_) and ammonium metavanadate (NH_4_VO_3_) were added in a stoichiometric ratio of 1:2 to the 40 mL of distilled water in a small beaker and then stirred. Next, 0.03 g of urea was added to the solution, and the resultant solution was transferred to a Teflon autoclave after 2 h of stirring and heated at 180 °C for 18 h. The resultant green-colored product was centrifuged with water/ethanol and washed several times. Finally, ErVO_4_ was dried at 80 °C for 24 h to be used for further experiments (Figure 1).

### 2.4. Synthesis of Graphitic Carbon Nitride (g-CN)

The thermal treatment method was followed for preparation. A quantity of 5 g of g-CN of melamine was added to 25 mL distilled water and stirred at 90 °C, and the resultant precipitate of melamine–cyanuric acid (MCA) was dried until there was no moisture in the sample. Then, it was transferred to N_2_ inserted tubular furnace, and calcined at 550 °C for 3 h. Yellow powder was collected for the preparation of a composite [32]. 

### 2.5. Preparation of ErVO_4_@g-CN-Modified GCE

A composite was prepared by changing the weight ratio of ErVO_4_ and g-CN in H_2_O, followed by sonication for 15 min. For fabrication, the glassy carbon electrode (GCE) was polished using alumina slurry and cleaned with water/ethanol to remove contamination. Then, this well-prepared GCE was kept for drying at room temperature for further usage. The optimized composite (weight ratio of 3:1, ErVO_4_: g-CN in H_2_O) was coated onto the GCE using the drop-casting method, which was then kept in a laboratory oven at 60 °C for drying [33].

## 3. Results and Discussion

### 3.1. Characterization of ErVO_4_@g-CN

Synthesized g-CN, ErVO_4_, and ErVO_4_@g-CN were analyzed with an X-ray diffractometer (XRD) as shown in Figure 1a. Peaks in the XRD spectrum of crystalline ErVO_4_ exhibit diffraction patterns at 2 theta values of 25.07° as well as at values of 31.56°, 33.67°, 35.75°, 40.72°, 45.17°, 48.42°, 49.92°, 51.46°, 58.07°, 62.91°, 64.96°, 70.80°, and 74.52°, which correspond to the (200), (211), (112), (220), (202), (301), (103), (321), (312), (400), (420), (332), (204), (224), and (512) planes, respectively (JCPDS 01-072-0860). The g-CN shows an amorphous nature in XRD, which can be seen in the composite of the ErVO_4_@g-CN/GCE pattern near the 2θ values of 27°. These results confirm the formation of a crystalline ErVO_4_ nanocomposite. 

The ErVO_4_@g-CN compound was further analyzed with FTIR for bond formation details, as shown in Figure 1b. The sharp peak at 533.38 cm^−1^ describes the Er-O stretching vibrations, and the V-O is represented by the stretching vibrations at 997.61 cm^−1^. The peak at 1011.20 cm^−1^ corresponds to the stretching vibration of (V=O). The peaks at 1401.56 cm^−1^ and 1634.91 cm^−1^ are due to the stretching vibrations of (C-N) and C=N in g-CN. The stretching vibrations of (N-H) in −NH_2_ and =N−H can be defined from the broad peak at 3381.80 cm^−1^. The confirmed s-triazine ring of the bending vibration is at 810.39 cm^−1^ [34,35,36,37]. The tetragonal crystal system with a 141/amd space group of ErVO_4_ is shown in Figure 1c. Er^3+^ is attached to eight equivalent O^2−^ atoms, and V^5+^ is bonded to four equivalent O^2−^ atoms.

Coral patterns, such as the nano-sized ErVO_4_-attached g-CN, can be seen in the Figure 2a FE-SEM images. A further examined ErVO_4_@g-CN microstructure from the HR-TEM images is displayed in Figure 2b,c. Nano-sized ErVO_4_ can be seen on sheet-like g-CN from the HR-TEM images in Figure 2b; the lattice fringe with an interplanar spacing value of 0.32 nm corresponding to the (200) crystal plane of ErVO_4_ is exhibited in Figure 2c. Also, the SAED image of Figure 2d reflects the crystal plane from the XRD analysis. The elemental distribution of compounds from the EDS analysis was studied, and this analysis further proved the presence of elements. The atomic percentages of C, N, O, V, and Er are 35.8%, 34.1%, 18.2%, 6.9%, and 5.1%, respectively, as depicted in Figure 2e. Furthermore, EDS mapping from the TEM images confirms the presence of erbium, vanadium, oxygen, carbon, and nitrogen, as shown in Appendix A.

### 3.2. Electrochemical Analysis in [Fe (CN)_6_]^3−/4−^ Solution

EIS can determine electrical responses between electrolytes, the interface of modified electrodes, and redox reactions, which helps us to further understand the system’s mass transfer, charge transfer, and diffusion processes. The Nyquist plot is designed with a real part (Z′) and an imaginary part (Z″). The Nyquist plot provides frequency-dependent resistance data equivalent to Randles’ circuit. R_s_, C_dl_, R_ct_, and Z_w_ are the resistance of solution (supporting electrolyte), double layer capacitance at the electrode surface, charge transfer resistance, and the Warburg impedance in the circuit, as seen in Figure 3a. The prepared modified electrode was immersed in a solution of 0.1 M KCl containing 5.0 mmol/L of [Fe(CN)_6_]^3−/4−^ with reference and counter electrodes. Resistance changes of bare GCE, ErVO_4_/GCE, g-CN/GCE, and ErVO_4_@g-CN/GCE are shown in Figure 3a, and the consecutive R_ct_ values are 321.59 Ω, 50.31 Ω, 58.90 Ω, and 32.42 Ω. As can be seen, the composite-coated modified electrode has the smallest semicircle due to the increase in electron transfer at the electrode surface with nano ErVO_4_@g-CN. 

Performance of the electrochemical process was continued in order to optimize the modified electrode. Like the EIS, the experiment was carried out in 0.1 M KCl containing 5.0 mmol/L [Fe (CN)_6_]^3−/4−^ at a scan rate of 0.05 V s^−1^ in potential between −0.2 and 0.6 V. As can be seen in Figure 3b, the anodic peak currents of bare GCE, ErVO_4_/GCE, g-CN/GCE, and ErVO_4_@g-CN/GCE were given consecutively at 92.87, 104.16, 85.65, and 114.34 µA, and the lowest potential difference of 0.09 V (peak separation, ∆E_p_ = *E*_p,a_ − *E*_p,c_) was obtained for ErVO_4_@g-CN/GCE. Figure 3c shows the CV in different scan rates obtained from 0.02 to 0.2 V s^−1^ solution with 5 mM [Fe (CN)_6_]^3−/4−^ in 0.1 M KCl. The peak currents increase with an increasing scan rate. Figure 3d displays a corresponding linear graph of I*_p_* vs. (scan rate)^½^. The good linearity of the graph implies the diffusion control process of the electrochemical reaction. The Randles–Sevcik equation (Equation (1)) is used to calculate the electrochemically active surface area (EASA) of the electrode [38].
I_p_ = 2.69 *×* 10n^3/2^*AD*^1/2^*Cʋ*^1/2^(1)
where, 

*I_p_* = Redox peak current

n = Number of electrons participating in the reaction (n = 1)

A = EASA

D = Diffusion coefficient (7.6 × 10^−6^ cm^2^ s^−1^)

C = Concentration of [Fe (CN)_6_]^3−/4−^ (0.005 M)

ʋ = Scan rate (0.05 V s^−1^)

A better reversible reaction can be obtained in the presence of the ErVO_4_@g-CN composite, and this modified electrode can contribute an electrochemically active surface area (EASA) of 0.2 cm^2^ to the system.

### 3.3. Electrochemical Analysis

#### 3.3.1. Electrochemical Behaviors toward DMZ Detection

Dimetridazole (DMZ) is a derivative of the nitroimidazole drug that treats bacterial infections in veterinary medicine. Thus, the need for a sensitive and effective sensor for the detection of drugs which cause allergies is essential. Carbon-based composite modified electrodes have received more attention in recent years due to their high sensitivity and selectivity. Before carrying out the experiment in the presence of DMZ, the current responses for bare GCE, ErVO_4_/GCE, g-CN/GCE, and ErVO_4_@g-CN/GCE in 0.1 M PBS (pH = 7) were analyzed (Appendix A), followed by the electrochemical responses for all electrodes toward 200 µM of DMZ and of PBS (pH 7.0) by cyclic voltammetry with a scan rate of 0.05 V s^−1^, as shown in Figure 4a,b. The reduction peak potential of modified ErVO_4_@g-CN/GCE had a highest recorded value (20.89 µA) at −0.68 V, producing a better-defined peak than the bare electrode. The efficiency of a material-coated electrode was further examined with different coating of ErVO_4_@g-CN/GCE (2, 4, 6, 8, and 10 µL), by dropcasting method as shown in Figure 4c,d; the 8 µL-coated electrode was modified for further electrochemical studies. 

#### 3.3.2. Effect of pH

The influence of different pH values (3, 5, 7, 9, and 11) on current was investigated in the presence of 200 µM DMZ (scan rate 50 mVs^−1^) for ErVO_4_@g-CN/GCE. It can be seen in Figure 5a,b that I_p_ increases from pH 3 to 7 and decreases from pH 7 to 11. The data show that E_pc_ was shifted toward a negative potential with an increasing pH (7–11) due to low current density hydrogen ions. It is shown that a maximum reduction current of −20.91 µA was obtained when pH 7 was used in the medium, as indicated by the equal numbers of proton and electron transfers in the reaction; for this reason, pH 7 can be used for any further analyses [39].

#### 3.3.3. Effect of Concentration and Scan Rate

By changing concentration, performance can be optimized for further electrochemical performance. Figure 5c shows the cyclic voltammogram for a modified ErVO_4_@g-CN/GCE electrode current response with changes in the concentration of 50 to 400 µM of DMZ in 0.1 M PBS with pH 7 at a scan rate of 50 mV s^−1^. Current values increased linearly with increasing DMZ concentration as shown in Appendix A. The linear relation of current vs. concentration is I = −0.04649C − 8.72 (R^2^ = 0.9831). Also, a reduction peak current was studied for different scan rates from 20 to 200 mV s^−1^, as shown in Figure 5d. Negative current peaks were linearly increased when analyzing the increasing scan rates, indicating a faster charge transfer. A linear regression equation of I = −0.6807C − 8.833 with R^2^ = 0.9941 in current vs square root of scan rate graph in Appendix A, conveys that DMZ is under a diffusion-controlled process. 

#### 3.3.4. Differential Pulse Voltammetry (DPV) Study for Determination of DMZ and Real Sample Analysis

The DPV method has a higher accuracy of sensitivity for electrochemically active substances. This experiment used a 0.1 M PBS in potential between −0.1 V and 0 V, a pulse amplitude of 0.05 V, a pulse time of 0.5 s, and a scan rate of 50 mV s^−1^ throughout the addition of DMZ from 0.5 µM to 863.5 µM. Figure 6a,b show the DPV performance and corresponding calibration plot. I_pa_ increases linearly, and the linear regression equation is y = −0.0195 x −4.5369 (R^2^ = 0.9931). The limit of detection and sensitivity are calculated from using Equations (2) and (3) below [40]. Here, calculated values of the LOD and sensitivity are 0.001 µM and 2.523 µA µM^−1^ cm^2^, respectively.
(2)$$\mathrm{Limit}\;\mathrm{of}\;\mathrm{detection}=\frac{3\times Standard\;deviation}{Slope}$$
(3)$$Sensitivity=\frac{Slop\;value\;of\;low\;concenctration}{Electrode\;surface\;area}$$

The modified electrode of ErVO_4_@g-CN/GCE was applied in real samples for the determination of DMZ. Milk was collected from a shopping mart for the DPV analysis. Then, 10 µL of milk was injected into 10 mL of PBS and centrifuged; a known amount of 0.001 M of DMZ was then added. For further experiments, an analysis of the solution was carried out using previously optimized conditions. The resultant current response and calibration plot are shown in Figure 6c,d. The response obtained from the DPV shows that this system can detect DMZ in these real samples, providing good results at potential at −0.683 V; the LOD value was 0.008 µM for the real sample.

#### 3.3.5. Selectivity, Reproducibility, Repeatability and Stability

We studied the interference of co-existing substances in the DMZ sample and inspected for selectivity of the modified electrode. Current response to the addition of interference compounds is presented in Figure 7a as well as in the relative bar diagram of Figure 7b. The studies for the ErVO_4_@g-CN/GCE electrode was performed by adding 20 µM of ascorbic acid, uric acid, glucose, K^+^, Zn^+2^, and Ca^+2^ each in the presence of 0.01 M DMZ. It was observed that the ErVO_4_@g-CN/GCE-modified electrode showed good current response toward DMZ and had no significant interferences from co-interfering compounds.

The experiment of repeatability and the respective bar diagram for the modified electrode in DMZ was used six times, as can be seen in Figure 8a,b, and in the same condition as before; the RSD value for the six trials is 2.01%. The reproducibility of ErVO_4_@g-CN/GCE was investigated for three different electrodes in 0.01 M of 100 µL DMZ, as shown in Figure 8c; the corresponding bar diagram is shown in Appendix A. The above-mentioned test gave a relative standard deviation (RSD) of 2.97% and, thereby, along with the obtained RSD values of reproducibility and repeatability, indicates a good performance for the DMZ sensor. Furthermore, the stability of DMZ was evaluated for 60 cycles in the presence of 0.01 M of DMZ and pH 7 phosphate buffer solution; Figure 8d displays the result. The current response of modified ErVO4@g-CN/GCE has good stability in the DMZ environment [41,42].

## 4. Conclusions

In this research study, we successfully developed nano-sized ErVO_4_ via the hydrothermal method for a modified ErVO_4_@g-CN/GCE-based electrochemical sensor for detecting DMZ. The composite was examined with XRD, EDS, and HR-TEM, and the confirmed crystal structure was discussed. The rare earth metal vanadate with a two-dimensional structure modified electrode contributed modified electrodes that revealed excellent electrocatalytic activity with respect to individual and bare electrodes. Furthermore, ErVO_4_@g-CN/GCE resulted in a lower limit of detection of 0.001 µM toward DMZ in the wide working range of 0.5 to 863.5 µM by the differential pulse voltammetry method. This ErVO_4_@g-CN-coated modified electrode system reported a highly active surface area, high conductivity, and less charge transfer resistance. Furthermore, the proposed ErVO_4_@g-CN/GCE sensor showed good selectivity toward DMZ compared with six different interfering compounds (ascorbic acid, uric acid, glucose, K^+^, Zn^+2^, and Ca^+2^) from DPV electrochemical analysis, and good cycle stability was also observed for 60 cycles. Finally, it was found that the above-discussed electrochemical sensor shows promise in being able to recognize DMZ in actual samples. 

## Data Availability

Data is contained within the article or Appendix A.

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
