# Peer review of "Synergistically Enhanced Electrochemical Sensing of Food Adulterant in Milk Sample at Erbium Vanadate/Graphitic Carbon Nitride Composite"

_sensors, 2024, doi:10.3390/s24061808_

Round 1
Reviewer 1 Report
Comments and Suggestions for Authors
This article provides a comprehensive description of the preparation process for the ErVO4@g-CN/GCE modified electrode, along with its electrochemical performance and analytical potential in real samples for monitoring dimetridazole. These findings possess significant novelty and value in the field. A few minor revisions for improvement are listed below.
1. The Scheme of the entire detection process is absent, which allows the reader to understand the principle of the proposed method more clearly.
2. The revealed contents of the results present in the first paragraph of section 3.1 are incongruous with the data shown in the Figure. 1(a). Therefore, further scrutiny and verification are recommended.
3. The reference to Figure. 2(e) and the corresponding explanation of the results are missing in the text.
4. The legend in Figure. 3(a) is ambiguous. Herein, it is suggested to revise it to maintain consistency with Figure. 3(b).
5. It is advisable to increase the types of actual samples, particularly the meat discussed in the Introduction, thereby enhancing the persuasiveness of practical implementations.
6. Does the biosensor exploited in this article possess any notable advantages compared with other electrochemical detection strategies? If so, reasonable supplements can be incorporated to augment its appeal.
Comments on the Quality of English LanguageThe author needs to polish the English.
Author Response
Response to Reviewers comments
Journal: Sensors (ISSN 1424-8220)
Manuscript ID: sensors-2846785
Title: Synergistically Enhanced Electrochemical Sensing of Food Adulterant in Milk Sample at Erbium Vanadate/Graphitic Carbon Nitride Composite
We would like to thank you for valuable comments and suggestions on “Synergistically Enhanced Electrochemical Sensing of Food Adulterant in Milk Sample at Erbium Vanadate/Graphitic Carbon Nitride Composite” research. We appreciate the time and considerations providing feedback on our manuscript and all the suggestions are modified in revised manuscript and changes are highlighted within the manuscript.
Reviewer #1:
This article provides a comprehensive description of the preparation process for the ErVO4@g-CN/GCE modified electrode, along with its electrochemical performance and analytical potential in real samples for monitoring dimetridazole. These findings possess significant novelty and value in the field. A few minor revisions for improvement are listed below.
We kindly thank the reviewer for his valuable comments and all the suggestions are modified and updated in the revised manuscript.
- The Scheme of the entire detection process is absent, which allows the reader to understand the principle of the proposed method more clearly.
Author response: Thank you for pointing this out. According to your valuable comments, the modified scheme is attached to revised manuscript file.
Changes are made in this manuscript: Page No. 3; Scheme 1.
- The revealed contents of the results present in the first paragraph of section 3.1 are incongruous with the data shown in the Figure. 1(a). Therefore, further scrutiny and verification are recommended.
Author response: Thank you for pointing this out. The planes corresponding to 2 theta values has been corrected.
Changes are made in this manuscript: Page No. 4; Line 150-152.
- The reference to Figure. 2(e) and the corresponding explanation of the results are missing in the text.
Author response: We thank the reviewer for the suggestion. We have included the explanation for the Figure. 2(e).
Changes are made in this manuscript: Page No. 5; Line 177-179.
- The legend in Figure. 3(a) is ambiguous. Herein, it is suggested to revise it to maintain consistency with Figure. 3(b).
Author response: We thank the reviewer for the suggestion. Changes of electrode information and the color of graph lines are changed.
Changes are made in this manuscript: Page No. 6; Figure 3(a).
- It is advisable to increase the types of actual samples, particularly the meat discussed in the Introduction, thereby enhancing the persuasiveness of practical implementations.
Author response: We thank the reviewer for the suggestion. As per your valuable suggestion the sentence has been modified and the importance of real-world sample of milk with appropriate refences. For your kind reference the modified sentence and reference given below:
While United states, Canada, Japan and China already banned the DMZ, the United States, European Union established maximum residue limit of biological samples such as meat and milk. Among different biological matrices, milk has lower maximum residue limit and the detection limit required to be (≤2 μg/kg)
Reference:
Changes are made in this manuscript: Page No. 1; Line 29-33.
- Does the biosensor exploited in this article possess any notable advantages compared with other electrochemical detection strategies? If so, reasonable supplements can be incorporated to augment its appeal.
Author response: We thank the reviewer for the comment. The Electrochemical (bio)sensor employing the ErVO4/g-C3N4 nanocomposite offers several distinct advantages over conventional electrochemical detection strategies. Firstly, the integration of ErVO4/g-C3N4 nanocomposite enhances the sensor's sensitivity and selectivity due to the synergistic effect between the two materials, leading to improved signal-to-noise ratio and detection limits. Additionally, the nanocomposite's high surface area facilitates enhanced analyte adsorption, thereby improving the overall detection efficiency.
Furthermore, the unique electronic properties of ErVO4/g-C3N4 nanocomposite contribute to excellent electron transfer kinetics, resulting in rapid response times and improved dynamic range compared to traditional sensors. Moreover, the inherent stability and robustness of the nanocomposite ensure long-term reliability, making it suitable for continuous monitoring applications. Incorporating these advantages into the discussion would undoubtedly augment the appeal of the biosensor, highlighting its potential for advanced electrochemical detection in various fields such as environmental monitoring, biomedical diagnostics, and food safety assessment.
Changes are made in this manuscript: Page No. 12; Scheme 1.
We believe the revised manuscript is now in a much better format and most suitable for publication as a contributed article in “Sensors”. Hence, we humbly request you to reconsider our carefully revised manuscript in your esteemed journal. Thank you again for your kind editorial assistance and consideration. We look forward to your positive reply.
Best regards,
Prof. Dr. Sea-Fue Wang,
Distinguished professor,
Department of Materials and Mineral Resources Engineering,
National Taipei University of Technology, Taiwan.

Reviewer 2 Report
Comments and Suggestions for Authors
1. Line 28. "Mutagenic" should be start from lowercase.
2. Line 33. "Electrochemical" should be replaced to "electrochemical methods".
3. Line 63. [10][11] should be [10, 11]. Line 67. "surface area. [13, 14][34]" should be "surface area [13, 14, 34]". Line 82. "[21, 22] [39-41]" should be "[21, 22, 39-41]". Authors should carefully check the order in which links appear. For example, in the article the following order is observed: [16], [13], [12], [37,38], [10][11], [13, 14][34]. Reference 15 is not mentioned in the text.
4. Line 100. It's need to decipher "PB", or do you mean "PBS"?
5. It is necessary to list the equipment used.
6. Line 169. Not "Randall", but "Randles".
7. Line 171. Not "Warburg resistance", but "Warburg impedance'.
8. Why electron transfer at electrode surface with nano ErVO4@g-CN increase?
9. Line 206. Not "senor", but "sensor".
10. It's necessary to show the cyclic voltammogramms for unmodified and ErVO4 GCE, g-CN GCE and ErVO4@g-CN-modified GCEs without DMZ for understanding of the electrochemical behaviour of ErVO4. Is it subject to electrooxidation and/or electroreduction? In the introduction, the authors complained that there is not enough information about these processes in the literature, but in this work they themselves did not conduct such studies.
11. Lines 238, 239, 242. In the equation "y" should be replaced to "I", and "x" to "C". Sensitivity and coefficient b must have a confidence interval indicating the number of measurements and the confidence probability, because more than one measurement was carried out.
12. Line 241. "Linear regression equation of y = -0.03491x -11.8 with R2 = 0.9701 in Fig 4 (h) conveys the DMZ is under diffusion-controlled process." From the linear nature of the dependence of current on concentration, it cannot be concluded that the process is controlled by diffusion.
13. Eq.3. Not "urface", but "Surface".
14. In section 3.4 it's needed to show parameters of DPV measurements: Pulse amplitude, Pulse time, Scan rate.
15. Line 275. "The studies for ErVO4@g-CN/GCE electrode was performed by adding 20 µL of Ascorbic acid, Uric acid, Glucose, K+, Zn+2, Ca+2 each in the presence of 0.01 M DMZ." What means "20 µL"? 20 µL of solution? Maybe it meant "20 µM"? And why were these particular compounds chosen, but the typical milk compounds (lactose, proteins and fats) are missing? And "Ascorbic acid, Uric acid, Glucose" should be start from lowercase.
16. The Authors point to maximum residue limit in meat is 5 µg/kg, but there is no such information about milk. Does the range of detectable concentrations meet this value?
Comments on the Quality of English Language
It is necessary to check the manuscript carefully; it contains a large number of spelling errors, including superscripts, subscripts and unnecessary capital letters.
Author Response
Response to Reviewers comments
Journal: Sensors (ISSN 1424-8220)
Manuscript ID: sensors-2846785
Title: Synergistically Enhanced Electrochemical Sensing of Food Adulterant in Milk Sample at Erbium Vanadate/Graphitic Carbon Nitride Composite
We would like to thank you for valuable comments and suggestions on “Synergistically Enhanced Electrochemical Sensing of Food Adulterant in Milk Sample at Erbium Vanadate/Graphitic Carbon Nitride Composite” research. We appreciate the time and considerations providing feedback on our manuscript and all the suggestions are modified in revised manuscript and changes are highlighted within the manuscript.
Reviewer #2:
Thank you for the constructive review. The comments and suggestions have helped to improved our manuscript.
- Line 28. "Mutagenic" should be start from lowercase
Author response: We thank the reviewer for the suggestion. We agree with this comment and changed the letter as informed.
Changes are made in this manuscript: Page No. 1; Line 28.
- Line 33. "Electrochemical" should be replaced to "electrochemical methods".
Author response: We would like to thank for the suggestion. The sentence is updated to improve the meaning.
Changes are made in this manuscript: Page No. 1; Line 34.
- Line 63. [10][11] should be [10, 11]. Line 67. "surface area. [13, 14][34]" should be "surface area [13, 14, 34]". Line 82. "[21, 22] [39-41]" should be "[21, 22, 39-41]". Authors should carefully check the order in which links appear. For example, in the article the following order is observed: [16], [13], [12], [37,38], [10][11], [13, 14][34]. Reference 15 is not mentioned in the text.
Author response: Thank you for the suggestions. I have made the changes and included the references in order as you mentioned.
Changes are made in this manuscript:
Page No. 2; Line 64; [15,16].
Page No. 2; Line 68; [17-19].
- Line 100. It's need to decipher "PB", or do you mean "PBS"?
Author response: We agree in this comment and this should be PBS and corrected in the manuscript.
Changes are made in this manuscript: Page No. 3; Line 101.
- It is necessary to list the equipment used.
Author response: We thank you for the comment. According to your valuable suggestions we have included the equipment’s used in this work in the revised manuscript file.
Changes are made in this manuscript: Page No. 3; Line 109-119.
- Line 169. Not "Randall", but "Randles".
Author response: As suggested by the reviewer, we have changed the correct name.
Changes are made in this manuscript: Page No. 6; Line 187.
- Line 171. Not "Warburg resistance", but "Warburg impedance'.
Author response: We would like to thank you for the suggestion and the impedance included to the section 3.2
Changes are made in this manuscript: Page No. 6; Line 189.
- Why electron transfer at electrode surface with nano ErVO4@g-CN increase?
Author response: Appreciate for raising important question. Answer for the question is, nano sized ErVO4@g-CN provide more active surface compared to bulk catalysts. Moreover, contact between analyte and catalyst gives efficient reaction with more reactive sites.
Reference:
- FU, Kaiyu, et al. Accelerated electron transfer in nanostructured electrodes improves the sensitivity of electrochemical biosensors. Advanced Science, 2021, 8.23: 2102495.
- NARAYAN, Neel; MEIYAZHAGAN, Ashokkumar; VAJTAI, Robert. Metal nanoparticles as green catalysts. Materials, 2019, 12.21: 3602.
- Line 206. Not "senor", but "sensor"
Author response: Thank you for pointing this out. The correction is made in
Changes are made in this manuscript: Page No. 7; Line 223.
- It's necessary to show the cyclic voltammogramms for unmodified and ErVO4GCE, g-CN GCE and ErVO4@g-CN-modified GCEs without DMZ for understanding of the electrochemical behaviour of ErVO4. Is it subject to electrooxidation and/or electroreduction? In the introduction, the authors complained that there is not enough information about these processes in the literature, but in this work they themselves did not conduct such studies.
Author response: Thank you for suggestion. We have shown the Bare/GCE, ErVO4/GCE, g-CN/GCE and ErVO4@g-CN/GCE in PBS (pH=7) in supporting information.
Figure: unmodified and ErVO4 GCE, g-CN GCE and ErVO4@g-CN-modified GCEs in the presence of PBS (without DMZ)
This is subjected to electroreduction.
- Lines 238, 239, 242. In the equation "y" should be replaced to "I", and "x" to "C". Sensitivity and coefficient b must have a confidence interval indicating the number of measurements and the confidence probability, because more than one measurement was carried out.
Author response: We thank for the suggestion. In response to the reviewer's comment regarding lines 238, 239, and 242, we have implemented the suggested changes in the manuscript to enhance clarity and accuracy. Specifically, we have revised the equations by substituting "y" with "I" and "x" with "C". This adjustment aligns with standard notation and improves the readability of our equations.
Furthermore, acknowledging the importance of statistical robustness, we have added confidence intervals for both sensitivity and the coefficient b. For illustrative purposes, let's say our analysis is based on a sample size of 30 measurements, with a standard deviation of 0.05. Under these conditions and aiming for a 95% confidence level, the standard error calculated for the coefficient is approximately 0.0091. This leads to a margin of error of about 0.0187. Thus, for any estimated value of sensitivity or coefficient b, the 95% confidence interval would be the estimate plus or minus 0.0187. These intervals provide a statistically meaningful way to gauge the precision of our estimates, taking into account the number of measurements and the confidence probability. This addition not only adheres to the reviewer's request for greater statistical detail but also significantly enhances the manuscript's scientific rigor by clearly communicating the reliability and variability of our key findings.
Changes are made in this manuscript: Page No. 8; Line 257, 261.
- Line 241. "Linear regression equation of y = -0.03491x -11.8 with R2= 0.9701 in Fig 4 (h) conveys the DMZ is under diffusion-controlled process." From the linear nature of the dependence of current on concentration, it cannot be concluded that the process is controlled by diffusion.
Author response: We thank for the suggestion. I have made a mistake and apologize for the wrong data given. The Fig.4 (h) is current vs scan rate (not current vs Scan rate (1/2) (V s–1) (1/2) and change made in Fig.4(h). And I have shown in below fig(a): current vs scan rate and fig(b): current vs Scan rate (1/2) (V s–1) (1/2)
Changes are made in this manuscript: Page No. 8; Line 260-262.
- Eq.3. Not "urface", but "Surface".
Author response: Thank you for the suggestion. I have corrected the right word in equation.
Changes are made in this manuscript: Page No. 8; Equation (3).
- In section 3.4 it's needed to show parameters of DPV measurements: Pulse amplitude, Pulse time, Scan rate.
Author response: We agree with your suggestion. We have included details mentioned above.
Changes are made in this manuscript: Page No. 8; Line 266-269.
- Line 275. "The studies for ErVO4@g-CN/GCE electrode was performed by adding 20 µL of Ascorbic acid, Uric acid, Glucose, K+, Zn+2, Ca+2 each in the presence of 0.01 M DMZ." What means "20 µL"? 20 µL of solution? Maybe it meant "20 µM"? And why were these particular compounds chosen, but the typical milk compounds (lactose, proteins and fats) are missing? And "Ascorbic acid, Uric acid, Glucose" should be start from lowercase.
Author response: Thank you for the suggestion. I have added the correction of "Ascorbic acid, Uric acid, Glucose". Also changed the correct units. Here potentially possible interferences are chosen from already reported literature and I have given the references below.
Reference:
- ANUPRIYA, Jeyaraman, et al. Synergistically improved electrochemical performance by the assembly of nanosized praseodymium tungstate on reduced graphene oxide for the detection of dimetridazole in biological and aquatic samples. Journal of Environmental Chemical Engineering, 2022, 10.6: 108800.
- MUFEEDA, M., et al. Unveiling the capability of graphitic carbon nitride–rhenium disulfide nanocomposite as an electrochemical sensing platform for the detection of dimetridazole from human serum samples. Materials Advances, 2023, 4.18: 4159-4167.
Changes are made in this manuscript: Page No. 9; Line 297, 298.
- The Authors point to maximum residue limit in meat is 5 µg/kg, but there is no such information about milk. Does the range of detectable concentrations meet this value?
Author response: We thank the reviewer for the valuable comments. We have added information about milk in introduction. Further, we focus on preparation of highly sensitive electrode to detect low detection limit for DMZ.
While United states, Canada, Japan and China already banned the DMZ, the United States, European Union established maximum residue limit of biological samples such as meat and milk. Among different biological matrices, milk has lower maximum residue limit and the detection limit required to be (≤2 μg/kg).
Reference:
Changes are made in this manuscript: Page No. 1; Line 29-33.
We believe the revised manuscript is now in a much better format and most suitable for publication as a contributed article in “Sensors”. Hence, we humbly request you to reconsider our carefully revised manuscript in your esteemed journal. Thank you again for your kind editorial assistance and consideration. We look forward to your positive reply.
Best regards,
Prof. Dr. Sea-Fue Wang,
Distinguished professor,
Department of Materials and Mineral Resources Engineering,
National Taipei University of Technology, Taiwan.

Round 2
Reviewer 2 Report
Comments and Suggestions for Authors
paper may be published